

# Application of multiplet structure deconvolution to extract scalar coupling constants from 1D NMR spectra

Damien Jeannerat[1,2], Carlos Cobas[2]

[1]NMRprocess.ch, Geneva, 1200, Switzerland
[2]Mestrelab Research S.L. Santiago de Compostela, A Coruña, 15706 Spain

*Correspondence to*: Damien Jeannerat (damien.jeannerat@protonmail.com)

**Abstract.** Multiplet structure deconvolution provides a robust method to determine the values of the coupling constants in first-order 1D NMR spectra. Functions simplifying the coupling structure for any spins and for doublet with unequal amplitudes were introduced. The chemical shifts of the coupling partners causing mild second-order effects can, in favourable
cases, be calculated from the slopes measured in doublet structures. Illustrations demonstrate that deconvolution can straightforwardly analyse multiplet posing difficulties to humans and, in some cases, extract coupling constants from unresolved multiplets.

## 1 Introduction

When studying organic compounds, 1D [1]H NMR spectra are often the only, and sufficient, analytical method engaged. All
chemist known how to describe simple multiplet structures. It consists in determining their position (chemical shift), integral and, when possible, their coupling structure by identifying doublets (*d*), double doublets (*dd*), *etc*. Any serious analysis also includes the values of the scalar coupling constants. Indeed, the multiplet structures provide important topological and structural information on organic compounds and natural products. For example, the presence of a methyl group on a carbon bearing a proton will produce a quartet structure, the conformation of a double bond has clearly distinct geminal coupling
constants, dihedral angle influences vicinal coupling constants (Karplus, 1963), *etc*.
These NMR parameters are not only helping researchers to identify their products, but also allow reviewers to assess the validity of the argument supporting their identification and benefit the community by providing very precious information when structurally similar compounds are encountered. It is therefore of fundamental importance to provide chemists with the most powerful tools to analyse multiplets in particular in the cases where the multiplet structure is too complex to be deciphered
by visual inspection because of high degeneracy (complex structures such as "*dqd*"), partial overlap of multiplet structures or second-order effects.
The measurement of scalar coupling constants has been the object of very intense academic work, either for direct application to 1D spectra (McIntyre and Freeman, 1992;del Río Portilla and Freeman, 1993) or involving the development of NMR pulses sequences producing spectra where coupling constants can be measured easily (Marquez et al., 2001) even in situations of
extensive overlap (Prasch et al., 1998;Kiraly et al., 2020;Berger, 2018). The sad observation is that the impact of these developments is extremely limited because the broad community of chemists almost exclusively rely on basic 1D [1]H spectra to extract coupling constants. Even if users of NMR were aware of them, they would be reluctant to use experiments going against the common practice in their fields except if they were producing key results - something NMR coupling constant alone, rarely do.
We shall report here on a significant improvement of the analysis of 1D [1]H spectra based on multiplet-structure deconvolution which will be introduced in a future release of the Mestrelab's Mnova software (Mnova NMR version 14.3.0, 2021). This method was originally developed during the '90s to automatize the analysis of multiplet structure from spectra generated by one of the forgotten pulse sequences: the soft-COSY experiment. (Emsley et al., 1990) Multiplet deconvolution shown to be applicable to the more "standard" DQF-COSY experiment (Jeannerat, 2000) after simply increasing the direct acquisition time
to provide higher resolution - something which was problematic because of the data storage available at that time, but is no





longer justifiable. We shall demonstrate the power of the application of multiplet deconvolution outside the world of 2D correlation spectroscopy, where positive and negative peaks coexist and cause specific challenges (Jeannerat and Bodenhausen, 1999) and apply it to the common – not to call it mundane – multiplet structure present in standard 1D NMR spectra.

## 1.1 Multiplet structure in 1D spectra

For weakly coupled spin systems, the multiplet structure observed in 1D spectra can be seen as the result of the combination (from left to right in Fig. 1) of the resonance frequency, the effects of the exponential relaxation, $B_0$-field inhomogeneity and scalar coupling ($J$) interactions.(Metz et al., 2000)

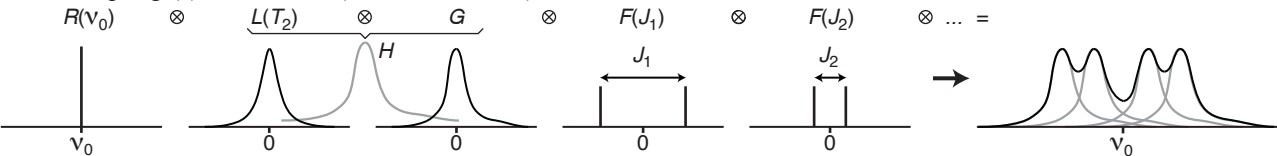

**Figure 1: Multiplet structure of a 1D NMR spectrum expressed as the convolution product of the contributions of the Larmor**
**frequency of the nucleus $R(v_0)$, the Lorentzian line shape $L(T_2)$, the $B_0$-field inhomogeneity $G(inhom.)$ and each scalar coupling constant $F(J_i)$.**

In the frequency domain, the $\otimes$ symbol stands for the "convolution product" which correspond, in the time domain, to pointwise multiplication. The detected FID is, indeed, the product of the chemical shift evolution, the exponential decay (producing a Lorentzian shape $L(v)$), a function reflecting the effect of $B_0$ inhomogeneity (often modelled as Gaussian shapes
in both domains) and cosine modulations caused by the first-order $J$-coupling interactions (resulting to doublet in the 1D spectrum).

In liquid state NMR and in isotropic media, mathematically, a doublet $F(J)$ is expressed by a pair of so-called $\delta$ functions, *i.e.* a function returning zero values everywhere except for the values of $v = -J/2$ and $+J/2$, where $J$ is the scalar coupling constant. When taking the convolution product of any set of $\delta$ functions with any other function, typically a spectral lineshape
$H$, the position and amplitude of the $\delta$ functions can be understood as to indicate where the convoluted function $H$, should be duplicated before summation (see the right part of Fig. 1).

The deconvolution consists in reverting the effect of the convolution of a given function. The symbols $\oslash F$ is sometimes used to represent the process reversing the effect of $\otimes F$. But what seems to be a simple division is misleading because, in reality, the $\otimes$ symbol stands for an integral which has no general inverse. One should, instead, look for an inverse of the $F$ function
and replace the deconvolution operator with the convolution with an inverse function $\otimes 1/F(J)$. An alternative is to operate in the time domain but the simplicity of the division is compensated by the difficulty of dealing with division by zero.

Whether it is applied in the time or frequency domain, deconvolution is a difficult process prone to produce noise and artifacts. The low efficiency of image deblurring is a reminder of this fundamental difficulty. In the field of NMR, methods improving the spectral lineshape (relaxation and field inhomogeneity) were developed in the group of Gareth A. Morris under the general
framework known as *Reference Deconvolution*, (Morris et al., 1997;Morris, 2002) but the difficulty to automatize them makes deconvolution disappointingly disregarded. We should not discuss, here, the deconvolution of spectral lineshape but concentrate on the deconvolution of the effect of coupling interactions called "multiplet-structure deconvolution". Following developments in the time domain (Le Parco et al., 1992;Bothner-By and Dadok, 1987;Prost et al., 2006) the frequency domain became more popular to avoid the back-and-forth Fourier transformation. This branch of research started in the '80s in the
group of R. Freeman (del Río Portilla et al., 1994) continued in the '90s in G. Bodenhausen's group (Huber and Bodenhausen, 1993a, b;Jeannerat, 2000;Jeannerat and Bodenhausen, 1999) and continued to resonate when the ACCA method (Cobas et al., 2005) was developed for Mestrelab's software Mnova.

This paper presents a follow-up of one published in 1999 (Jeannerat and Bodenhausen, 1999) which showed how to effectively use a set of inverse function of $F$ to obtain, through a recursive procedure (Novič and Bodenhausen, 2002), the list of the
coupling constants and the lineshape $G$. The inverse function $M(J^*)$, reminiscent of the set of $\delta$ functions seen in the '90s (del Río Portilla et al., 1994) is discussed in the next paragraph. While earlier work focused on the simplification of simple doublet, we will present applications to the deconvolution of structures originating from the coupling with partners with S > 1/2, and deal with partially overlapping multiplet structures. It is generally limited to the analysis of first-order multiplets but a method





to accommodate some "roof effect" is discussed (*i.e.* deal with the unequal amplitude of the two lines of doublets caused by
second-order effects, see §1.7) and turn it into an advantage to determine, in favorable cases, the chemical shifts of the coupling
partners.

**1.2 Deconvolution of doublet structures**

The process we shall use to reverse the effect of a doublet is the convolution product of the starting multiplet with the
simplification function $M(J^*)$ (see Fig. 2). The deconvoluted function will usually be an experimental multiplet with a possibly
complex structure, but the top of Fig. 2 shows how it applies to the model function of a doublet $F^{1/2}(J)$ to provide the
mathematical demonstration that $M$ simplifies any doublet structure.

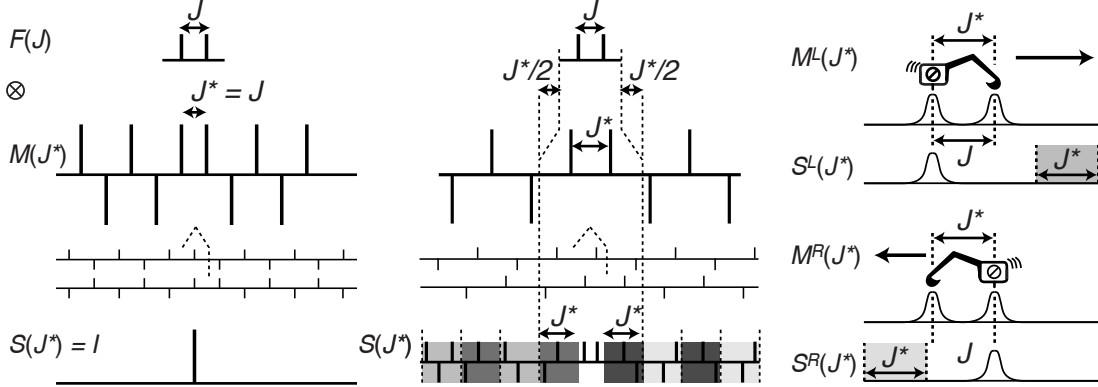

**Figure 2: (left) The function $M(J^*)$ cancels the doublet of a weakly coupled partner with a spin 1/2 when $J^* = J$. (middle left)
Duplicating $M$ and shifting one copy by the distance $J$ indeed cancels all except the central peak resulting in the expected
simplification (bottom left). (centre) When $J^* \neq J$, the convolution product results in the complex set of $\delta$ functions $S$ (middle
bottom). (right) Illustration of the algorithm starting at one side of the spectrum. It can be seen as walking from one boundary of
the multiplet, in the manner of an integrator, and subtracting at the distance $J^*$, the amplitude measure locally.**

The simplification of the doublet structure into the singlet $I$ is obtained when the distance between the $\delta$ functions of
$M(J^*)$ corresponds to the true coupling. Otherwise $S = F(J) \otimes M(J^*)$ consists in a complex set of $\delta$ functions that we call
deconvolution artifacts. Determining the coupling constants consists in measuring the degree of simplification of the result $S$
as a function of $J^*$. In principle, $S$ are infinite arrays, but beyond the central core (see the section framed by dotted lines in the
center of Fig. 2) two patterns (in grey) repeats themselves with alternating signs.
It has been demonstrated (Jeannerat, 1997;Bothner-By, 1995) that the result of the application of the arrays of $\delta$ functions
$M(J^*)$ is equivalent to a recursive process starting at either side of the multiplet illustrated on the right part of Fig. 2. It shows
that the calculation can be understood as the result of moving a cursor (represented by the excavator) over the vector of signal
amplitudes and subtract, at the arm's distance, representing the tested value of $J^*$, the value measured at its current position.
After chopping off the margins which should be empty (in grey) to align the sub-multiplets, the sum of $S^L$ and $S^R$ is
numerically equal to $S$. This approach has the advantage of being computationally quite effective: each point in the
deconvoluted spectrum is calculated as the sum of only two points. When $J^* = J$, the remote subtraction eliminates the second
occurrence of the substructure split by $J$. A key property of this algorithm is that artifacts and the residuals of sub-multiplet
subtraction increase from the starting boundary to the other as the number of operations increases. It is tempting to reduce
artifacts by taking only the most favorable half of the simplified multiplets, but it is not generally recommended because it
often produces a small discontinuity in the middle of the resulting multiplet. The most significant advantage of the side-to-side
process is to facilitate the measure of the similarity of $S^L$ and $S^R$ to test the success of the simplification process. It takes
advantage of presence of deconvolution artifacts indicating that $J^* \neq J$ only in the right, respectively left marginal regions (in
grey in Fig. 2 ) of $S^L$ and $S^R$. Instead of the classical Chi-squared test, we favored a scalar product (Huber and Bodenhausen,
1993a) as measure of similarity but this should only be a matter of personal preference. Finally, the presence of two sub-



multiplets makes it possible to simply discard one of them when an artifact (solvent singlet, spike, *etc.*) or partial overlap makes the complementarity fail. (see §1.8)

## 120 1.3 Recursive simplification of multiplet structure

As mentioned earlier, deconvolution produces artifacts because the components of multiplets are never perfectly identical. This causes imperfect multiplet subtraction due to the presence of random noise, *etc*. These artifacts tend to be duplicated for each $\delta$ function of $M$ and their presence is often a limiting factor for the analysis of complex structures. Reducing them is therefore decisive to make multiplet deconvolution reliable. Deconvolution with large values of $J^*$ generate fewer artifacts
because they contain less $\delta$ function per unit distance. This is one of the reasons to aim first at the largest coupling present in any multiplet. Note that it is the opposite to what a human would do based on the knowledge that only the smallest coupling can be readily measured between the first and second outermost lines of any multiplet. The other reason to start with the largest coupling constants is that values of $J^* = J/(2n + 1)$ (where $n \in \mathbb{N}^*$) also produces multiplet structure with no marginal signal (see the extremum at $J^* = J/3$ in Fig. 3) making it unreliable to use symmetry as a criterion for the determination of values
smaller than one third of the largest coupling of the multiplet. This problem becomes irrelevant when using a recursive process where the result of the simplification of the largest coupling constants is used as the starting point of the next step. In order to limit the accumulation of artifacts, applying symmetry at each step or taking the best part of the sub-multiplet $S^L$ and $S^R$, is generally recommended.

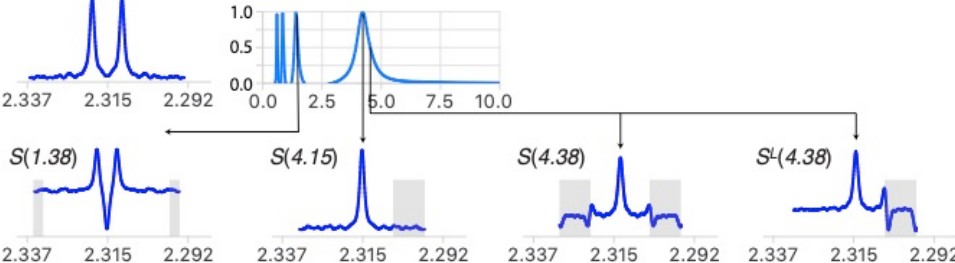

**Figure 3: Multiplet structure obtained by convolution of the top left doublet with the $M$ function for $J^* = 1.38$ ( $= J/3$), 4.15 ( $= J$) and 4.38 ( $= J + 0.23$) Hz and $M^L$ for 4.38 Hz.**

An alternative to the measure of symmetry consists in testing the sum of the absolute values of the resulting multiplets. It should reach half the value of the starting multiplet when it is simplified. It requires only $M^L$ and $M^R$ (instead of both for the comparison using symmetry) but we used it solely when only one side was available, for example when analysing partially
overlapping multiplets (see §1.8).
Having a good starting multiplet is also important and requires the spectrum to present a reasonable baseline and the identification and subtraction of solvent and other artifact peaks by the processing software prior to the deconvolution process increases the chances of success of multiplet deconvolution. On the other hand, random noise, should only impact on the confidence level of the identification of each coupling.

## 145 1.4 Multiplet lineshape

An interesting property of multiplet deconvolution is to make no assumption on the underlying signals lineshape. It can be any mixed Lorentz/Gauss function, transformed by any type of apodization or lineshape deconvolution. Multiplet deconvolution only needs a pattern, of any shape, to be equally split. Figures 4 and S1 illustrate this point for a classical phase distortion, a distorted lineshape, and the results of a resolution enhancement replacing the multiplet with its second derivative (Wahab et
al., 2018). This being said, the above-mentioned symmetrization procedure applied at each step of the simplification process is generally beneficial (compare Figs. S2 and S3). It averages out noise, artifacts and asymmetrical coupling structures caused by mild second-order effects (roof effects) but it usually does not work properly when signals do not have a pure absorption lineshape (*i.e.* when the phase is incorrect) or when the $B_0$ inhomogeneity produces a severely asymmetrical lineshape.




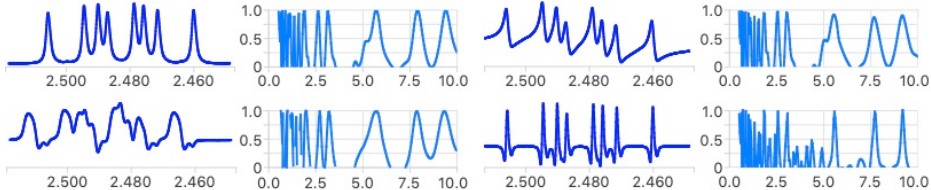

**Figure 4: Comparison of the first step of the analysis of reference multiplet before (top left) and after introduction of a phase error (top right), a distorted line-shape (bottom left) and using its second derivative (bottom right). The measured coupling constants are the same in all cases as the position of extrema in the four error functions shows. The narrower extrema of the error functions in the latter indicate a higher potential to identify small coupling constants. See Fig. S1 for a more detailed analysis.**

**1.5 Spin system degeneracy**

When spin systems are $n$-fold degenerated, the convolution with $M(J^*)$ is simply applied $n$ times before measuring the degree of simplification. A trivial example of quartet structure is shown in Fig. S4. A user-controlled specification of degeneracy simplifies the structure in one step and avoids the possible errors of the automatic identification which takes as *degenerate* coupling constants differing by less than 0.5 Hz.

In highly degenerate systems, the outermost signals may be missed when using automatic analysis. A post-processing procedure using multiplet simulations (see §2.3) to test if higher levels of degeneracy match better the starting multiplet turned out to be useful in the case illustrated in Fig. 5.

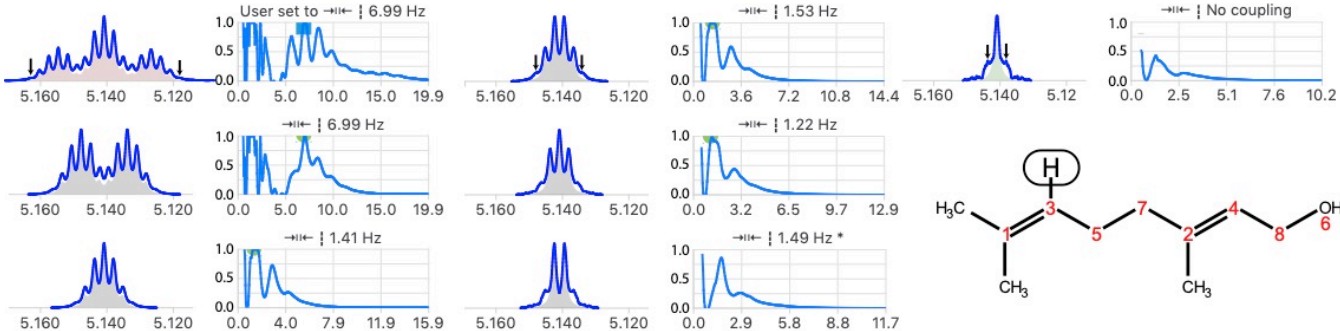

**Figure 5: Example of automatic analysis failing to identify a triplet structure after six steps of simplification. Because the outermost lines (see arrows) were too small. It identified a problem because the reconstructed multiplet (triple quintet filled in red) was not perfectly matching (the scalar product was < 0.99) the starting multiplet (in blue). The post-analysis procedure found a satisfactory match for a triplet (7.0 Hz) caused by methylene (C5) of heptet ($^4J$ = 1.3 Hz) caused by the coupling with two methyl groups bound to C1 of geraniol.**

**1.6 Coupling partner with S > 1/2**

When compounds contain deuterium ($S = 1$, 1:1:1 structure), boron ($^{11}$B, $S = 3/2$, 1:1:1:1 structures) and other S > 1/2 nuclei, the simplification of the multiplet structure with $n$ equal-amplitude $\delta$ function is necessary when quadrupolar relaxation is not effectively hiding the coupling. The general inverse function $M(J^*)$, valid for $S \geq 1/2$, consists in a core of $n$ $\delta$ functions with unit values producing the main singlet. It is flanked, at both sides by a repeat of blocks made of one negative $\delta$ function with intensity equal to $-(n-1)$ and $n-1$ lines with unit intensities. The function $M^{3/2}(J^*)$ is shown in Fig. 6.





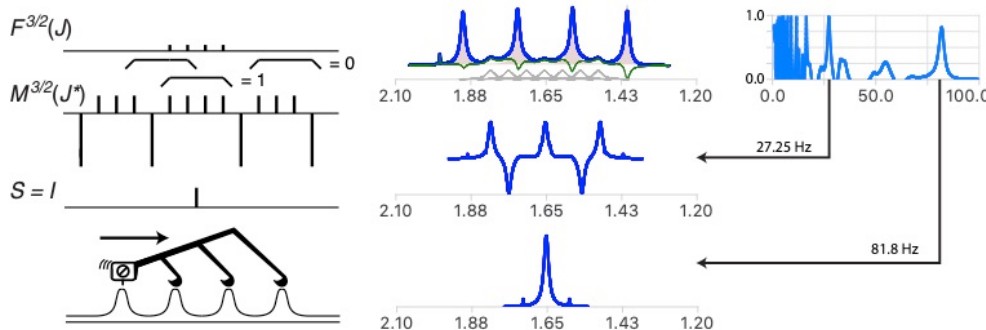

**Figure 6: (top right) Model multiplet structure for coupling with a $S = 3/2$ partners. (middle right) Representation of the series of $\delta$ functions $M^{3/2}(J^*)$. The braces above show examples of integration and illustrate that it is zero for all positions except the central one. The side-to-side process is quite similar to the case of $S = 1/2$ except that subtraction of the cursor position has to be made to the $n - 1$ positions located $J^*$ further in the data array. Analysis of the proton spectrum of BH$_4^+$. Specifying the spin of the partner to be $3/2$ replaces the wrongly assigned $dd$ ($J = 163.6, 81.6$ Hz) with the correct 1:1:1:1 quartet with $J = 81.6$ Hz. The presence of the minor multiplet for the $^{10}$B isotope (20% natural abundance, $S = 3$) with a $\gamma$ one third that of $^{11}$B is highlighted in grey at the top. Note the artefact (suspected to be due to traces of acetone) was scaled down by symmetrization in the last structure (bottom).**

The convolution product of $F^{3/2}(J)$ and $M^{3/2}(J^* = J)$ is indeed $I$ because the sum of any set of four consecutive $\delta$ functions ($\sum_{i=m}^{m+n} \delta_i$) is equal to zero for all positions except the middle one. The side-to-side recursive functions $M^{L,S>1/2}(J^*)$ and $M^{R,S>1/2}(J^*)$ also exist (see the bottom left of Figure 6).

For a given coupling constant, the density of $\delta$ in the simplifying function is the same as for spin 1/2, but the fact that the average of the intensities of the $M$ functions increase with spin order and the fact these multiplets are more extended should generally make the simplification of complex multiplets more prone to artifacts.

The analysis of the main 1:1:1:1 quartet of the proton of borohydride produced by the coupling with $^{11}$B is straightforward (see Fig. 6). The process is not disturbed by the presence of the signal of the 20% abundance of the $^{10}$B isotopologue. Note that a refinement of the parameters involving a fit of the experimental spectrum with spectral simulation taking into account the presence of the two isotopes, could provide values for the isotopic shift and the line widths of both multiplets.

The analysis of a system including the deuterium atoms of the CHD$_2$ residual signal of DMSO-D6 is shown in Fig. S5.

### 1.7 Mild second-order effects

Because second-order effects are often observed in otherwise well-behaved multiplet (*i.e.* when only the peak intensities are uneven with no significant shift or additional second-order transitions) we introduced, in this work, the function deconvoluting doublets with non-equal amplitudes. Mild second-order effects are usually not impairing the extraction of coupling constants, especially when symmetrizing multiplet structures, but the validation of the results matching simulation with the experimental multiplet (see §2.3) may fail because of the mismatch of the peak amplitudes. Accounting for the roof effect restores high confidence in the results of the analysis. The simplification function $M^{1/2}(J^*, \theta)$ (see Fig. 7) includes the strength of the coupling as a parameter expressed a $\theta = \tan^{-1}(J/\Delta\delta)$ where $\Delta\delta$ is the difference in the chemical shift of the two coupling partners expressed in Hz. It produces the identity function when taking the convolution product with the second-order doublet $F^{1/2}(J, \theta)$ where the ratio of peak amplitudes

$$r = \frac{1 - \sin(\theta)}{1 + \sin(\theta)}. \tag{1}$$

Note that the stronger the coupling, the more artifacts will be amplified at one side of the resulting multiplet. In order to avoid having two independent variables to adjust, $\theta$ can be optimized only for values of $J^*$ corresponding to an extremum of the values obtained for $\theta = 0$.



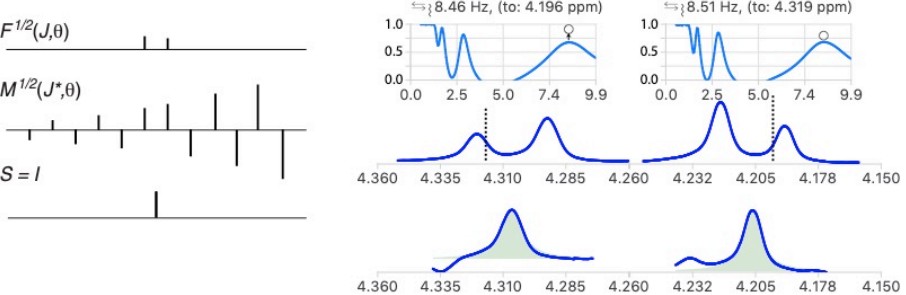

**Figure 7: (left) Illustration of the model function of a doublet with non-equal amplitude $F^{1/2}(J,\theta)$ and its inverse function $M^{1/2}(J^*,\theta)$. The amplitude ratio of the absolute values of two consecutive $\delta$ in $M$ is the inverse of the ratio in $F^{1/2}(J,\theta)$. (right)**
**Analysis of AB multiplet of a sample of a 300 MHz spectrum of taxol. $\Delta\delta/J$ = 6.6, $2\theta$ = 8.6 deg. and the ratio of signal amplitudes $r$ = 0.74. The error in the determination of partner chemical shift was about 0.01 ppm. The grey dots indicate the level of the measure of the quality function after optimization of $\theta$.**

Besides the fact that taking into account the roof effects increases the level of confidence that any measure splitting is correct, exploiting them provides, when it can be measured with sufficient precision, the chemical shift of the relevant coupling partner.
(Sykora, 2008) This information normally requires a 2D COSY spectrum, but extracting this information for the 1D spectrum has the additional advantage to identify which of the coupling observed in a given multiplet corresponds to the designated coupling partner. In other words, instead of only qualitatively pointing to the side of the spectrum where one should look for the coupling partner of a signal showing roof effects, one can actually point to the partner according to:

$$\delta_{partner} = \delta_{ref.} + \frac{J}{\tan(\theta)}. \tag{2}$$

Such a determination of the coupling partner position is expected to provide useful values for a relatively narrow range second-order effects of say $3 < \Delta\delta/J < 20$, an example of which, shown in Fig. 7 was taken from the [1]H spectrum of a sample of taxol. (Peng et al., 1997) Indeed, the second-order effect should not be too strong to avoid the minor line of the doublet to be too small to be measured accurately. Similarly, if the effect is too weak, the difference of intensity of the two lines of the doublet becomes too small to be significant and generate large errors because $1/\tan(\theta)$ becomes quite large when $\theta$ is small (see the
example in Fig. S6). Taking into account the error on the partner position by standard error propagation calculation is therefore essential to avoid misinterpretation. Obviously, the chances of success of simply picking the nearest multiplet about the expected chemical shift as true partner depends on the complexity of the spectra.

**1.8 Partially overlapping multiplets**

The side-to-side deconvolution allows to deal with multiplets that are partially superposed. Figure 8 shows that signal overlap over a distance smaller than the largest coupling present in the multiplet can be analysed successfully.
When attempting to simplify the left multiplet, the deconvolution process should be run from left to right and the deconvolution process stopped at the position where the integral reaches 1/2 in the reconstructed multiplet.



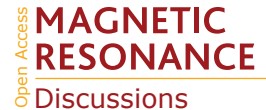


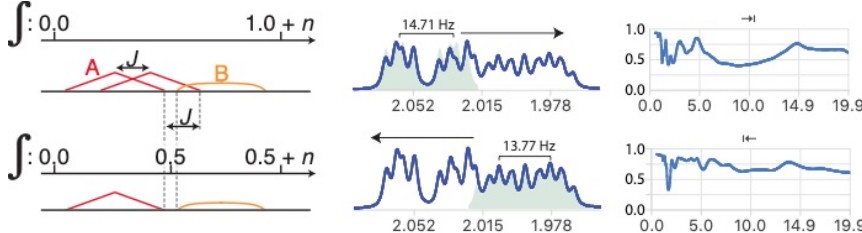

**Figure 8: Analysis of the multiplet A in the presence of a partially overlapping multiplet B with relative integral of 1 and *n* respectively. When deconvolution is successful, the left-to-right deconvolution process eliminates the second occurrence of the sub-multiplet A which effectively eliminates overlap (bottom left). (right) Example of separation of two slightly overlapping multiplet structures from a sample of artemicin dissolved in CDCl₃. The coupling constants *J* = 14.71, 4.77, 2.89 and 13.77, 2.55, 2.08 and 2.93 Hz were extracted from the left and right multiplets respectively and used to reconstruct the matching green multiplets. Note the slight mismatch due to second-order effects which could be ignored in this analysis. See Fig. S7 for more details of the simplification process.**

When testing different values of $J^*$, the measure of the success of the simplification cannot be applied on the whole spectral
region because the segment where the sub-multiplet subtraction occurs may be occupied by multiplet B. But successful deconvolution produces a minimal and predictable integral when $J^* = J$. This separation method requires to know the integral of the analysed multiplet relative to the total integral of the region of interest. This is usually not problematic as integration is generally straightforward. An example of analysis resulting to a sub-multiplet separation is shown on the right part of Figs. 8 and S7.
Following the analysis of A, the multiplet B can be analysed independently (from right to left) only if its largest coupling is also large enough, as in the example on the right of Fig. 8. Otherwise, a recursive process analysing and subtracting multiplet structures sequentially is necessary. This would impose the quality of the subtraction of each multiplet to be high enough to leave no significant residual signals on the remaining, but allows, in principle, to deal with more than two overlapping multiplets.
For more severely overlapping structures, the methodology developed to separate 2D multiplets (Jeannerat and Bodenhausen, 1996) could be adapted to the processing of 1D spectra.

## 2 Implementation

### 2.1 Deconvolution parameters

As mentioned above, the simplification of structure is a recursive process where the noise and artifacts tend to increase along
the course of the analysis and depend on the quality of the multiplet (complexity, base line distortion, presence of spurious peaks, *etc.*)
In order to increase the robustness of multiplet deconvolution, different set of parameters driving the deconvolution process are tested and their results validated (see next paragraph). These parameters are: the threshold for considering a deconvolution to be successful, the threshold for considering the recursive process to be completed, the range of values tested for the roof
effect, whether symmetry is applied before starting the analysis, whether symmetry is applied after each simplification, whether the baseline offset if corrected, whether the crude spectrum is used instead of the GSD-based synthetic spectrum (Cobas et al., 2008;Schoenberger et al., 2016), *etc*.

### 2.2 Limit of deconvolution analysis

Obviously, the recursive simplification stops when the result is a singlet. But the ability of multiplet deconvolution to find
unresolved coupling makes that simply testing the presence of only one extremum in the result of a simplification step could miss unresolved coupling constants (see §3.2). Testing values down to 1 Hz seems to be a good compromise for standard 1D



[1]H spectra. Note that sub-linewidth coupling constants should be used with great care and only to facilitate the process of a structure elucidation process. They should not be used as sole argument for any key interpretation. Indeed, the behavior of the deconvolution has not been studied sufficiently to provide a general and safe limit as to how far out it can be carried. A rigorous
analysis of unresolved structures would involve a process taking, among other elements, the lineshape found in other multiplet into account. When a candidate structure is available, predictable long-range coupling constants should be considered. Depending on the priority of the application, one may favor safety over ambition and decide to ignore any unresolved structure in a "fully automated" mode. Pushing the limit of multiplet analysis could be a user-triggered operation providing visualization tools to assess the validity of the results and correct them if necessary.

**2.3 Validation and post-processing**

The validation and ranking of the results of the different methods consists in the measure of the similarity of the starting multiplet with a reconstructed multiplet build using the extracted data. The latter are obtained by starting with the final singlet shape and reintroducing the measured coupling constants including, when relevant the tilt caused by second-order effect, and testing the similarity with the initial multiplet. In this work, the starting lineshapes were synthetic Lorentzian but allowing
more degrees of freedom (*i.e.* using Voigt, Generalized Lorentzian, *etc*. lineshape) should improve the match with the shape obtained at the end of the deconvolution.
Finally, a refinement of all parameters: signal intensity, chemical shift, coupling constants, but also roof effect, signal phase, baseline level, and lineshape parameters could be used to further improve the quality of the validation and take into account context-dependent information such as the degeneracy of the coupling constants.

**3 Results and discussion**

Multiplet deconvolution has been applied to numerous test spectra. Many multiplet structures which could not be analysed using the GSD peaks-based multiplet analysis (reported as "*m*") were correctly deciphered. It also has a lower occurrence of false positive – the situation where coupling constants are automatically extracted but turn out to be incorrect – thanks to the strict validation process. We shall only present here two typical examples where multiplet deconvolution succeeds while
manual analysis would be difficult.

**3.1 Human unfriendly multiplets**

Figure 9 shows a typical example of multiplet that cause difficulties to human analysis. The partial overlap of three transitions (see arrows) makes it difficult to interpret. The destiny of such a structure is to be called the bad name "*m*" for a lack of courage and proper tool to identify it. Multiplet deconvolution straightforwardly identify the largest coupling as 9.9 Hz. Similarly, the
seemingly triplet of the second step would have been assigned to a *t* with two equal coupling constants of 5.27 Hz - possibly causing confusion if nothing explains this apparent degeneracy. Here again, deconvolution has no difficulties to resolve overlap and identifies a double doublet with 6.32 and 4.22 Hz - almost 2 Hz difference for a final line width of 2.7 Hz.

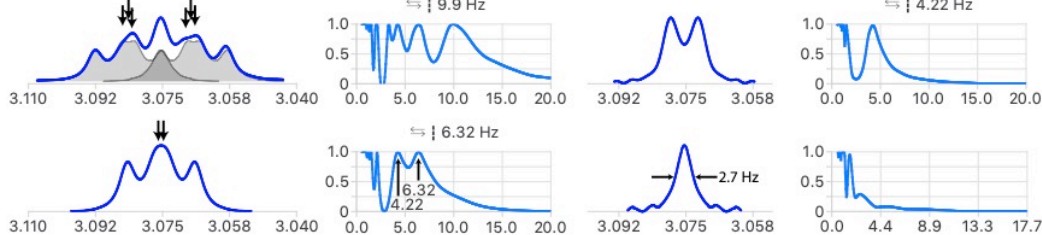

**Figure 9: Example of a *ddd* structure analysed straightforwardly by multiplet deconvolution. The ┊ symbol indicate that symmetrization was applied before deconvolution. The ⇆ symbol indicates that the results of the deconvolution running from both sides were added (instead of taking the best part of both of each of only the result of one direction).**





An example of successful analysis of multiplet structure which was too complex to be analysed manually is shown in Fig. S8.
The six coupling interactions, among which three were degenerate, produced 32 lines with no visible splitting patterns.


**3.2 Unresolved couplings**

In carbohydrate chemistry, geminal coupling constants are often too small to be resolved making it difficult to assign their
signals. The analysis of the multiplets of Fig. 10, identified a common coupling of 1.2 Hz which increases the confidence in
their geminal relationship. In the signal at 3.655 ppm, neither the 2.2 nor the 1.2 Hz were resolved in the starting multiplet
providing a good example of the potential of multiplet deconvolution.

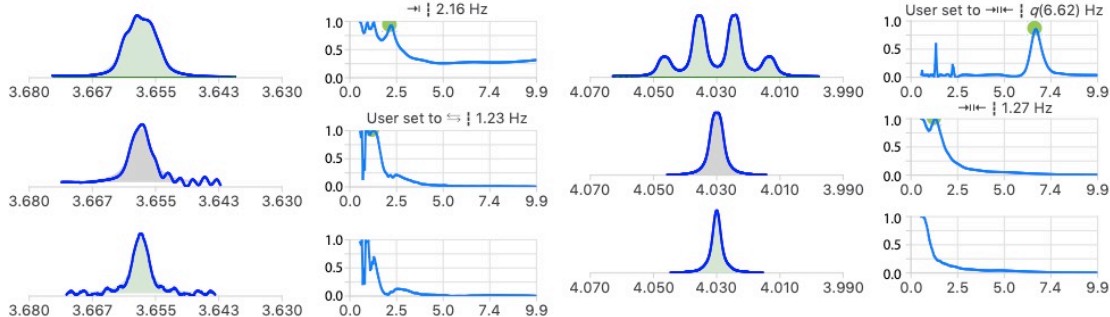

**Figure 10: Extraction of unresolved coupling constants of the proton signals of C4 (left) and C5 (right) of a 6-deoxy-pyranose unit.
The ⇥ symbol indicates that symmetrization was not applied and deconvolution run from left to right.**

**Conclusion**

The recursive analysis of NMR spectra by multiplet deconvolution demonstrated its ability to extract coupling constants in a
robust and fully automated manner. The results are validated when simulations based on the extracted data match the
experimental multiplet structures. Additional features including the analysis of structures produced by coupling partners with
$S > 1/2$ and of regions presenting signal overlap are proposed with the option of using a graphical interface providing a full
user control on the stepwise analysis of multiplet structures.
By reducing the number of the unsuccessfully analysed multiplet, multiplet deconvolution increases the information content
of NMR spectra at a much lower cost than increasing the field strength of NMR spectrometers. Combined with other efforts
aiming at automatizing the analysis of NMR spectra, these methods can significantly increase the quantity and quality of NMR
data available to the community. But this will only occur if researchers make their NMR spectra and extracted data available
as supplementary data deposited on public databases instead of providing only crude images of spectra and other non-computer
readable information.

**Data Availability**

The software presented here is proprietary of Mestrelab but the code for the recursive deconvolution of first-order multiplet
with spin 1/2 partners is available as a JavaScript node modules on GitHub https://cheminfo.github.io/multiplet-analysis.
The JCAMP-DX files of the spectra presented in this paper are available on Zenodo doi: 10.5281/zenodo.4616665. They can
be visualized with the open-source NMR displayer https://cheminfo.github.io/nmr-displayer and analysed by deconvolution
by selecting a multiplet with the mouse while pressing the shift key.





**Acknowledgements**

Damien Jeannerat acknowledges Milan Bochud and Théodora Quiriconi for comments as well as Luc Patiny and co-workers for the cheminfo environment and assistance for the JavaScript implementation of multiplet deconvolution.

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
