# Peer review of "Application of multiplet structure deconvolution to extract scalar coupling constants from 1D NMR spectra"

_Magnetic Resonance, 2021_

## Referee Comment (RC2)

[referee-annotated manuscript omitted]

---

## Author Response (AR1)

The reply to reviewers are https://mr.copernicus.org/#AC1 and https://mr.copernicus.org/#AC2

They are pasted below.
* * *
https://mr.copernicus.org/#AC1

Indeed, the series of delta function should be understood as being infinite. We changed the figure to make it more clear by adding dotted lines at both sides of the series of delta functions.

[Figure]

Concerning canceling antiphase structure (something we did not discuss in the paper, I would have used a different represenation (see right part of the image below... and after a step of change of sign). Note this figure is just shown for this disucussion and does not apear in the paper). But these model all have their limits - as the different understanding of the original figure demonstrates!

[Figure]

We also changed the legend of the figure to clarify the deconvolution process.
* * *
https://mr.copernicus.org/#AC2

**A: author statements, C: my comments A2 reply of Author1 in boldface**

A: On page 1, section 30 the author writes: "The sad observation is that the impact of these developments is extremely limited because the broad community of chemists almost exclusively rely on basic 1D 1 H spectra to extract coupling constants."

C: Indeed, in most cases chemists measure the couplings for reporting NMR spectra of their compounds to provide a "fingerprint" type information where there is no need for particularly accurate J-coupling information, and we can't blame them for not spending time on learning how to obtain such information with great precision and accuracy. Of course, if more accurate information is made available at their fingertips they will welcome the new tools.

**A2. No blame, indeed, software should provide chemists with J values.**

A: On page 2, section 65 the author writes: "In the field of NMR, methods improving the spectral lineshape (relaxation and field inhomogeneity) were developed in the group of Gareth A. Morris under the general 70 framework known as Reference Deconvolution, (Morris et al., 1997;Morris, 2002) but the difficulty to automatize them (i.e. automate the process) makes deconvolution disappointingly disregarded. "

C: I would argue that the proposed technique is likely meet a similar fate as the reference deconvolution. Of course, this does not mean that such techniques should not be developed because clearly there are situations where they make a difference. However, there are always limits of applicability (e.g. SNR) and a price to pay.

**A2: Future will tell if multiplet deconvolution meets the same fate. But the fact that it will replace the standard j-coupling measurement of MNova software allows to be optimistic.**

A: The author continues: "We should not (i.e. shall not) discuss, here, the deconvolution of spectral lineshape but concentrate on the deconvolution of the effect of coupling interactions called "multiplet-structure deconvolution".".

C: Perhaps it is worth mentioning that in many situations combining both techniques may, in fact, improve the results (see e.g. section 150).

**A2: We were distinguishing deconvolution of the lineshape v/s multiplet structrure. We consider them as two separate techniques, but yes, I agree that combining processing techniques is the best.**

A: Second-order effects.

C: The author only considers the symmetry ("roof") effects. However, there are other second-order effects, for instance the multiplet components may also have unequal linewidth due to various relaxation effects (quadrupolar, CSA etc.). How will this affect the analysis?

**A2: Deconvolution assumes equal lineshape in all transition. No, we did look at diffential relaxation effects.**

A: On page 9, section 295 the author writes: "Finally, a refinement of all parameters: signal intensity, chemical shift, coupling constants, but also roof effect, signal phase, baseline level, and lineshape parameters could be used to further improve the quality of the validation and take into account context-dependent information such as the degeneracy of the coupling constants."

C: While discussing refinement and validation options the author never considered simulations (e.g. for limited/localized spin systems), which should provide the ultimate means of verification.

**A2: Yes, it would be very usefull indeed.**

A: On page 10, section 335 the author writes: "Combined with other efforts aiming at automatizing (i.e. automating) the analysis of NMR spectra, these methods can significantly increase the quantity and quality of NMR data available to the community. But this will only occur if researchers make their NMR spectra and extracted data available as supplementary data deposited on public databases instead of providing only crude images of spectra and other non-computer readable information."

C: Surely the chemists will argue that this will occur when the spectra analysis becomes a pushbutton operation. Likewise, the readers of this article will surely agree that there is still a long way to go before this is achieved by analysing only 1D proton spectra. Clearly, some of the problems, mentioned in this article, such as estimating the position of the coupling partners, can be simplified by involving some very basic 2D data. Such a possibility should be at least mentioned.

**A2: We agree. Our call to chemists to share NMR spectra is independant of the quality of automated analysis. They should provide spectra (with or without) assignement - not the low-quality images pasted in word documents or pdf. This is the (somewhat out-of-the-focus of the paper) call.**

Generally, the manuscript is reasonably well written, the illustrations are of good quality, the references are appropriate. There are a few typos and phrasing issues, as suggested in the attachment. These, however, only require some minor corrections. Therefore, I am pleased to recommend the manuscript for publication subject to a minor revision.

**A2: Thank you very much for your comments and for the corrections!**